# Self-Renewal and Pluripotency in Osteosarcoma Stem Cells’ Chemoresistance: Notch, Hedgehog, and Wnt/β-Catenin Interplay with Embryonic Markers

**DOI:** 10.3390/ijms24098401

**Published:** 2023-05-07

**Authors:** Sara R. Martins-Neves, Gabriela Sampaio-Ribeiro, Célia M. F. Gomes

**Affiliations:** 1iCBR—Coimbra Institute for Clinical and Biomedical Research, Faculty of Medicine, University of Coimbra, 3000-548 Coimbra, Portugal; sara.neves87@gmail.com (S.R.M.-N.);; 2Institute of Pharmacology and Experimental Therapeutics, Faculty of Medicine, University of Coimbra, 3000-548 Coimbra, Portugal; 3CIBB—Center for Innovative Biomedicine and Biotechnology, University of Coimbra, 3000-548 Coimbra, Portugal; 4CACC—Clinical Academic Center of Coimbra, 3000-075 Coimbra, Portugal

**Keywords:** osteosarcoma, mesenchymal stem cell, cancer stem cell, self-renewal, Notch, Hedgehog, Wnt, pluripotency, SOX-2, KLF4

## Abstract

Osteosarcoma is a highly malignant bone tumor derived from mesenchymal cells that contains self-renewing cancer stem cells (CSCs), which are responsible for tumor progression and chemotherapy resistance. Understanding the signaling pathways that regulate CSC self-renewal and survival is crucial for developing effective therapies. The Notch, Hedgehog, and Wnt/β-Catenin developmental pathways, which are essential for self-renewal and differentiation of normal stem cells, have been identified as important regulators of osteosarcoma CSCs and also in the resistance to anticancer therapies. Targeting these pathways and their interactions with embryonic markers and the tumor microenvironment may be a promising therapeutic strategy to overcome chemoresistance and improve the prognosis for osteosarcoma patients. This review focuses on the role of Notch, Hedgehog, and Wnt/β-Catenin signaling in regulating CSC self-renewal, pluripotency, and chemoresistance, and their potential as targets for anti-cancer therapies. We also discuss the relevance of embryonic markers, including SOX-2, Oct-4, NANOG, and KLF4, in osteosarcoma CSCs and their association with the aforementioned signaling pathways in overcoming drug resistance.

## 1. Introduction

The presence of cancer stem cells (CSCs) within tumor tissues was first documented in 1937 by Furth and Kahn [1]. These authors inoculated a single mouse tumor cell in another recipient mouse and successfully generated a new tumor. In 1959 Makino proposed the term “tumor stem cells”, defining them as a minor subset of cells that were not responsive to chemotherapy and presented chromosomal alterations compared to the majority of cells in the tumor tissue [2]. Nevertheless, the first description of the “tumor stem cell” concept was later on proposed by Pierce and colleagues (1960–1988), owing to milestone experiments performed in mouse teratocarcinomas and squamous cell carcinomas [3,4,5]. The presence of cells in distinct differentiation stages and a probable hierarchical cellular organization were defined as “a concept of neoplasms, based upon developmental and oncological principles, states that carcinomas are caricatures of tissue renewal, in that they are composed of a mixture of malignant stem cells, which have a marked capacity for proliferation and a limited capacity for differentiation under normal homeostatic conditions, and of the differentiated, possibly benign, progeny of these malignant cells” [5]. Later on, in the 1970s, Metcalf and Moore also introduced the concept of leukemic stem cells, described as cells able to form colonies and self-replicative [6].

Both normal and tumor tissues possess a population of stem cells that operate key functions in the preservation of the structure and functionality of those individual tissues [7]. Stem cells possess a unique set of properties that differentiate them from generic cells, namely the ability to self-renew and also to generate phenotypically diverse cell types (differentiation). Self-renewal pertains to the inherent ability of the stem cells to originate two daughter cells, in which at least one of them conserves the potential to divide indefinitely without losing the stem cell-state properties, namely developmental capacity. Differentiation refers to the process in which an immature and unspecialized daughter cell acquires new individual properties and new functions. This so-called progenitor or transit-amplifying cell proliferates faster and is committed to those new functions within the tissues. In more recent decades, a wide array of studies and independent research groups has demonstrated that, in fact, virtually all cancer types are endowed with a subset of stem-like cells possessing these capabilities of normal stem cells, the so-called CSCs. This is also true for osteosarcoma, the most common primary malignant bone tumor [8]. The concept of CSCs’ existence in oncobiology has clearly evolved in recent decades and has been explored widely by different research groups. The reader is referred to recent reviews of outstanding quality that analyzed the role of, e.g., autophagy [9], epigenetic modifiers [10] such as DNA methylation [11], tumor microenvironment [12] and CSC’s niches [13], DNA repair signaling [14], and also studies that explored new attempts to develop targeted therapeutic strategies such as molecular docking [15], nanomedicine [16], and bioinformatic single-cell analysis [17]. The possible origin of CSCs in tumors remains a highly debatable topic [18,19].

We previously reviewed the recent literature concerning some of the signaling pathways that most contribute to resistance of osteosarcoma CSCs to conventional therapies, namely drug efflux transporters, aldehyde dehydrogenase activity, activation of survival-related pathways, adaptive metabolic routes, altered cell cycle and DNA repair, and enhanced apoptosis and tumor microenvironment modulation [8]. Herein, we propose to highlight some studies regarding the discussion around the osteosarcoma cell of origin and to revisit in more detail the activity and expression of pathways involved in self-renewal such as Notch, Hedgehog, and Wnt/β-catenin signaling and in cellular pluripotency (focusing the SRY-Box Transcription Factor 2 (SOX-2, *SOX2*), Homeobox protein NANOG (*NANOG*), Krueppel-like factor 4 (KLF4, *KLF4*), and Octamer-binding protein 3 (Oct-4, *POU5F1*) transcription factors), in an attempt to discuss the crosstalk that these pathways operate in osteosarcoma CSCs to further substantiate and mediate the resistance of CSCs to conventional chemotherapy.

## 2. Human Osteosarcoma—From Bedside (Clinical and Biological Observations) to Bench (Molecular Markers and Cell of Origin)

### 2.1. Incidence Patterns and Clinical Features of Osteosarcoma

Among the overall panel of primary cancers afflicting humans, bone tumors located in the skeleton are very rare. However, around 20–40% of all primary bone sarcomas are osteosarcoma, an aggressive tumor with clear predominance in children and adolescents (85% versus 15% in adults). This is a primary high-grade tumor, comprising around 4% of all childhood malignancies [20,21,22], afflicting mainly children up to 15 years (2.3% of all tumors) and adolescents (15–25 years, 2.6% of all tumors) [23,24]. The overall worldwide incidence of osteosarcoma is 2–3/million/year. In Europe, this incidence is particularly higher between 15 and 19 years, with a registered annual incidence peak of 8–11/million/year in Reference [25]. Between 1978 and 1997, the accumulated number of cases in this age group was 372 per million [26]. Information from the American Surveillance, Epidemiology, and End Results (SEER) database demonstrates a tripartite statistical pattern, with a peak during puberty (8.4–8.6 cases), followed by a plateau in middle age (1.7 cases) and a second peak in older adults (>60 years, 4.9 cases) [27]. Based on this particular high incidence of osteosarcoma at very young ages, the rapid bone cell turnover and growth typical of pubertal growth spurt are probably linked to the risk of developing osteosarcoma. Osteosarcoma prevalence varies with age in males and females [28]. Of note, below 15 years of age, the incidence is higher in females, but the ratio reverses to males after that age [27,28,29]. Overall, these numbers of incidence have been unchanged during the last 40–50 years [30,31], although an increase in incidence in individuals of Black ethnicity was recently identified [30].

Osteosarcoma localizes mainly in long bones of extremities near the most proliferative growth plates, namely the distal femur, proximal tibia, and proximal humerus. The most significant and common symptoms, depending on the location and velocity of tumor growth, include reduced joint movement, local pain, and swelling around the tumor mass [25]. Pain may be more intense with physical activity and during the night; moreover, patients may feel the pain firstly in an intermittent manner but with a tendency to be more persistent over time. Pain and swelling occur less frequently in adults compared to active, younger patients, which may retard a correct diagnosis. Moreover, misdiagnosis may occur quite often, with tendinitis and osteomyelitis being some types of confounding diagnoses [32,33].

The need to search for a differential diagnosis between osteosarcoma and several other bone-affecting diseases, namely chondrosarcoma, dedifferentiated chondrosarcoma, Ewing sarcoma, osteochondroma, bizarre parosteal osteochondromatous proliferation, fibrous dysplasia, osteoblastoma, and chondroblastoma, is of note, as vividly recommended by the recent WHO bone tumor classification [34]. The advent of artificial intelligence will probably have a positive impact in aiding the medical community to process radiological images and accelerate the accurate diagnosis of osteosarcoma [35], which may then help to bypass misguided surgeries and treatments [36].

### 2.2. Classification and Biological Features of Osteosarcoma

According to the World Health Organization (WHO), osteosarcomas are classified as osteogenic tumors that produce defective bone with osteoid deposition or bony matrix (Figure 1). The diverse types of osteogenic tumors are classified by criteria using a combination of radiography, histopathological, and microscopic image analysis. Altogether, these data uncover the multicellular complexity of osteosarcoma histological patterns, as summarized before [37,38].

High-grade surface osteosarcomas may also present variable amounts of cartilage and/or fibrous tissue, which are, in some instances, used to further sub-classify into fibroblastic, chondroblastic, or osteoblastic osteosarcoma [37]. There is no established correlation between prognostic significance and this tumor sub-classification, although some correlation between the distinct histological subtypes and specific clinical outcomes has been observed, especially when multi-agent therapy is used [39,40].

Osteosarcoma cells show a diversity of cytological features, with tumors presenting an anaplastic pleomorphic structure combining at least two types of cells: clear cells, epithelioid, fusiform, mono- or multinucleated giant cells, ovoid, plasmacytoid, small round cells, or even spindle cells [41]. Factors such as axial location, a large tumor volume, and the presence of metastatic lesions are related to a negative prognosis for patients. Moreover, elevated levels of alkaline phosphatase in serum [42] and tumors poorly responsive to neoadjuvant chemotherapy have a higher risk of development of metastasis and recurrent disease [43].

### 2.3. Osteosarcomagenesis and Associated Molecular Abnormalities

Conventional high-grade osteosarcoma shows a wide genetic instability that impedes the identification of unique tumor driver genes. Deletions and losses have been found in chromosomes 3q, 6q, 9, 10, 13, 17p, and 18q, and amplifications and gains in chromosomes 1p, 1q, 6p, 8q, and 17p. The less common types of osteosarcoma normally display genomic modifications dissimilar from those found in conventional tumors, and this information is clinically useful for establishing a differential diagnosis [44,45].

Genetic alterations located in the retinoblastoma (*RB1*) (chromosome 13q14.2) and *TP53* (chromosome 17p13.1) genes cause inactivation of the RB1 and p53 proteins, and are significant genomic modifications involved in osteosarcomagenesis, which may also contribute to metastatic disease. *TP53* and *RB1* disease-driving mutations have been identified in inherited familial syndromes with a predisposition to osteosarcoma, but these somatic genetic alterations are also very common in sporadic osteosarcoma [46]. Several genetic alterations that may play a key role in osteosarcoma development that were previously reviewed [45,47] are also outlined in Table 1.

### 2.4. Crosstalk between Mesenchymal Stem Cells (MSC) and Osteosarcoma

Osteosarcoma is regarded as a cell differentiation disease derived from the transformation of multipotent mesenchymal stem cells (MSCs). Defective osteogenic differentiation of MSCs has been linked to the development of osteosarcoma [63]. Some research groups have been modeling the initial benign developmental stages of osteosarcoma, in which altered MSCs lacking *CDKN2A* locus formed osteosarcomas in murine models [64,65]. These models have been used to search for initiating events in osteosarcomagenesis [66,67], and inaugurated a new research area attempting to discover the origins of osteosarcoma.

Multipotent MSCs appear in the mesodermal differentiation cascade of embryonic stem cells (ESCs) after the molecular specification of the three germ layers, endoderm, mesoderm, and ectoderm (Figure 2A). There is large evidence for an MSC origin of osteosarcoma based on the functional parallelism and phenotypic behavior observed between normal and transformed MSCs. In this model, the malignant transformation of MSCs may originate diverse types of bone and soft tissue tumors, including fibrosarcoma, chondrosarcoma, and liposarcoma [68] (Figure 2B). The transformation of malignant MSCs into osteosarcoma may be related to their osteoblastic differentiation program. In fact, the bone marrow stem cell niche and its associated secreted factors influence the MSC program within osteosarcomas [69], as will also be outlined. Furthermore, there is a clear parallelism between the differentiation cascade of normal stem cells (ESCs and MSCs) and the ability of CSCs to also originate “differentiated” cancer cells (Figure 2C).

Human osteosarcoma samples’ mesenchymal stem cells were shown to be genetically distinct from tumor cells, indicating that maintaining a unique niche may be essential for keeping osteosarcoma cells in an undifferentiated condition [70]. Human osteosarcoma samples with MSC-related features were shown to be genetically distinct from the remaining tumor cells, indicating that maintaining a unique niche may be essential for keeping osteosarcoma cells in an undifferentiated condition [71], which has also been demonstrated to encourage osteosarcoma growth and metastasis by activating STAT3 [72] and to induce doxorubicin resistance through JAK2/STAT3 signaling [73]. In their investigation of the interaction between normal and cancer cells’ metabolic programming, Bonuccelli et al. discovered that MSCs can increase lactate production in response to oxidative stress brought on by osteosarcoma cells, which in turn improves the tumor cells’ capacity for migratory growth [74].

Together, these findings show how interactions between tumor cells and the environment’s favorable microenvironment are critical for the survival and growth of tumor cells [75]. Still, intense research has been conducted in recent years on the mechanisms by which osteosarcoma cells exploit healthy MSCs, with a number of teams further investigating the signaling pathways that control MSC programs in osteosarcomas. It has been studied how p53 is involved in osteosarcomagenesis and how it functions abnormally in MSCs [76]. It is important to note that p53 status alterations can jeopardize bone homeostasis since they control crucial MSC osteogenic differentiation programs that, if disturbed, can result in the emergence of osteosarcomas. For instance, Rubio et al. demonstrated the origin of metastatic osteoblastic osteosarcoma from intrabone or periosteal inoculation of p53-deficient bone marrow or adipose tissue-derived MSCs. This osteoblastic osteosarcoma increased the expression of osteogenic markers in a Wnt signaling-dependent manner, and further increased the formation of the typical osteoid matrix deposited by osteosarcoma cells [77]. An alternative mechanism involves the family of proteins, so-called inhibitors of DNA binding (IDs). An ingenious investigation by Williams and colleagues found that the deubiquitinating enzyme USP1 conserved stem cell-like characteristics in osteosarcoma by stabilizing ID proteins and deubiquitinating ID proteins. Additionally, forced USP1 expression in MSCs stabilized ID proteins, prevented osteoblastic differentiation, and enhanced the proliferative phenotype. This research finds new potential treatment targets and implicates deubiquitination as a novel mechanism that may contribute to stem cell states in osteosarcoma [78]. Altmayer et al. found that human MSC induced to differentiate into adipocytes and osteoblasts for 7 days showed a percentage of cells within the culture with amplification of *CDK4* and *MDM2* [79], which are frequently amplified in osteosarcoma as outlined earlier in this text.

Signaling pathways involved in metastasis formation also seems to mediate the crosstalk between MSCs and osteosarcoma cells. Fontanella et al. suggested that bone marrow-derived MSCs co-cultured with U2OS cells lead to the activation of AKT/ERK signaling. Additionally, elevated CXCR4 levels in normal cells led to an increase in the migration and invasion of tumor cells. These authors utilized the novel CXCR4 inhibitor Peptide R for therapeutic purposes, which decreased cancer cells’ metastatic potential by impairing the crosstalk between tumor cells and normal stem cells [80]. Others demonstrated that conditioned medium from bone marrow-derived stem cells promoted the invasive and proliferative abilities of osteosarcoma cells, effects possibly mediated by the stromal cell-derived factor-1/CXCR4 signaling axis [81]. SDF1-CXCR4 axis also seems to be involved in the chemotaxis of MSC-derived exosomes loaded with doxorubicin to preferentially target MG63, HOS, and 143B cells over free doxorubicin [82]. Conversely, also osteosarcoma-derived extracellular vesicles exert epigenetic alterations in surrounding MSCs and expression of genes related to bone microenvironment remodeling, namely *MMP1*, *VEGF-A*, and *ICAM1* [83].

Very recent studies are exploring the role of microRNAs in the regulation of the crosstalk between MSCs and osteosarcoma cells using MSC-derived exosomes, which suggests a dual mediation. Indeed, depending on the type of microRNAs, those studies are uncovering both a tumor-promoting (e.g., miR-655 [84], miR-30c-5p [85], and miR-532-3p [86]) and tumor-suppressive (e.g., miR-206 [87], miR-150 [88], and miR-1913 [89]) modulation of cancer cell behavior in osteosarcoma. Altogether, these studies support the notion that microenvironment signals from surrounding MSCs are key contributors to osteosarcoma development. In fact, Sarhadi recently reviewed the complex network of interactions between osteosarcoma cells and MSCs mediated by a myriad of secreted extracellular vesicles and microenvironmental factors [75].

## 3. Targeting Signaling Pathways Involved in Cellular Self-Renewal

It is now widely known that CSCs can be found in various cancers. These cells exhibit numerous traits shared by adult and embryonic stem cells. Numerous studies have shown that CSCs frequently exhibit persistent activation or expression of parts of highly conserved signal transduction pathways involved in development, differentiation, and tissue homeostasis, such as the Notch, Hedgehog, and Wnt pathways. The tumorigenicity of CSCs appears to rely on the abnormal activity of the signaling pathways regulating the self-renewal and differentiation of stem cells. Additionally, many other signaling pathways that control cellular division, survival, and invasion interact with these embryonic pathways. Targeting Notch [90], Hedgehog [91], and Wnt/β-catenin [92] pathways represents a key tactic for preventing the self-renewal and spread of CSCs. This review provides an up-to-date summary of the latest findings on this topic [93,94] on osteosarcoma CSCs and presents novel insights into potential strategies to circumvent their inherent chemoresistance.

### 3.1. Notch Signaling Pathway

An essential developmental signaling route that regulates stem cell self-renewal is Notch signaling. Differentiation of somatic stem cells and crosstalk with other signaling pathways might also modify the net effect of Notch signaling [90,95]. When Notch transmembrane ligands (such as DLL1/3/4 and Jagged1/2) from one cell interact with transmembrane receptors (such as Notch1-4) from another cell, a proteolytic cascade is set off, which results in the release of an intracellular fragment that can interact with CSL transcription factors to control the expression of target genes such as *P21*, *CYCLIND1*, *CMYC*, and genes from the *HES* and *HEY* families. The active Notch intracellular domain (NICD) is cleaved by the enzyme γ-secretase in this signaling cascade (Figure 3). Due to its dual function as an oncogene if involved in the regulation of stem cell self-renewal or as a tumor suppressor if involved in the regulation of cellular terminal differentiation, there is a growing body of data linking alterations in the Notch signaling pathway with the development of cancer [96,97]. The Notch pathway has detrimental roles in cancer and other diseases, such as inflammatory disorders, and efforts have been made in the search for potential therapeutic drugs in both basic research and clinical trials, as recently reviewed [98].

The Notch pathway has been found to regulate CSCs in various malignancies, including medulloblastoma, glioblastomas, and pancreatic cancer [99]. Several studies suggest that the Notch pathway promotes resistance to both chemo- and radiotherapy, and the pharmacological inhibition of this pathway might be an interesting way to mitigate the ability of CSCs to bypass conventional therapies [95]. In fact, Notch inhibition has been shown to sensitize CSCs to radiation and cisplatin as well as to present activity against CSCs in mouse models. Many drugs targeting components of the Notch pathway have entered clinical trials and present great potential for overcoming the resistance of CSCs, particularly when combined with chemotherapy or other targeted agents [100]. Figure 3 provides an overview of the compounds that have demonstrated effective inhibition of Notch molecular players (adapted from [101]).

#### Notch Signaling in Osteosarcoma

Extensive research has been conducted exploring the contribution of Notch signaling to the regulation of osteosarcoma CSCs. Recent data indicate that the crosstalk between tumor cells and cells residing in the bone-marrow niche via direct contact and paracrine communication through soluble growth factors or extracellular vesicles contributes to apoptosis resistance via Notch activation [102]. The activation of the Notch pathway through its ligand Jagged1 has been related to an increased capacity of osteosarcoma cells to proliferate, be drug-resistant, and form metastasis [103]. In fact, *JAGGED1* is a direct target of *MIR-26a,* which exerts its tumor-suppressive effect through inhibition of the Jagged/Notch pathway [104]. Moreover, high expression of Jagged1 in clinical specimens has been associated with metastasis and recurrence of osteosarcoma, whereas Jagged knockdown reduced osteosarcoma cell migration and invasion, suggesting an oncogenic role for this protein in the development of osteosarcoma [103].

Tanaka et al. found a high frequency of tissue specimens overexpressing Notch receptors and ligands (*NOTCH2, JAGGED1*, *HEY1*, and *HEY2)*. Moreover, inhibition of Notch reduced in vitro cell proliferation, in vivo tumor formation, and cell cycle arrest, decreased expression of cell cycle promotors (e.g., cyclin D1), and increased expression of cell cycle suppressors (e.g., p21) [105]. Some authors reported a potential role of naturally occurring compounds such as cinobufagin [106] and oleanolic acid [107] in the induction of cell cycle arrest and apoptosis in osteosarcoma through Notch inhibition and the decrease in downstream signaling players. Further studies reinforce the crosstalk of Notch with key signaling that helps to mediate resistance to conventional therapies in osteosarcoma, such as hypoxia and *MRP1* [108], autophagy-related markers, such as the cysteine protease ATG4A [109], and key self-renewal signaling as Wnt/β-catenin seems mediated by upregulation of certain microRNAs, such as *MIR-135b* [110].

Paracrine and microenvironment-related signaling seem to be involved in the regulation of Notch in osteosarcoma CSCs. Recently, Zhuo et al. reported that interleukin-24 inhibited the expression of CD133^+^ cells and their tumorigenic capacity, mediated by the downregulation of both Notch and Wnt/β-Catenin signaling [111]. Additionally, co-culture of U2OS and 143B cells with exosomes derived from human umbilical vein endothelial cells enhanced the expression of CSCs markers *POU5F1* and *SOX2* and increased the proportions of STRO-1^+^/CD117^+^ cells. These effects were accompanied by upregulation of Notch1, Hes1, and Hey1, but were reversed by the γ-secretase inhibitor RO4929097 [112].

Doxorubicin and cisplatin have been reported as activators of the Notch pathway. For instance, at non-toxic doses, doxorubicin seems to inhibit the proliferation of osteosarcoma cells via the up-regulation of target genes such as *HEY1*, *NOTCH1*, *HES1*, and *HES5* [113]. Additionally, cisplatin at sub-lethal doses appears to select a subset of cisplatin-resistant cells displaying a mesenchymal cell profile and expression of postulated osteosarcoma CSC markers (STRO-1/CD117), a phenotype that was reversed by γ-secretase inhibition. Two other independent reports also explored the effects of exposure of osteosarcoma cell lines to low concentrations of doxorubicin [114] and cisplatin [115] and demonstrated an association to epithelial–mesenchymal transition (EMT) mediated by the upregulation of genes in the Notch signaling cascade, and increased expression of Notch receptors and target genes. These effects were also observed in vivo, including metastasis formation, and were counteracted by treatment with the selective γ-secretase inhibitor DAPT, which was also previously described as preventive of tumor recurrence in resistant xenograft tumors [116] and active against cisplatin-resistant osteosarcoma [117]. In this study, DAPT also depleted osteosarcoma CSCs, and the combinatorial treatment with cisplatin exhibited additive suppression on phosphorylated AKT and ERK survival-related pathways.

In vivo mouse models of osteosarcoma have proven to be useful in exploring the effects of Notch modulation in cell migration and metastasis formation. For instance, Mu et al. showed that *NOTCH1*, *NOTCH2*, *NOTCH4* genes and *HES1* and *STAT2* target genes had increased expression in the highly metastatic murine osteosarcoma K7M2 cell line, in comparison with the less metastatic murine K12 cells. Moreover, inhibition of Notch led to reduced ALDH activity in K7M2 cells [118]. Tao et al. established an osteosarcoma mouse model based on the conditional expression of NICD. The fundamental role of Notch activation in osteosarcoma was demonstrated in this model since forced *NICD* expression in immature osteoblasts was sufficient for the formation of tumors fully resembling the osteosarcoma pathophysiology. Moreover, the combination of Notch expression with p53 loss accelerated in a synergistic manner the development of osteosarcoma [119]. More recently, activation of the Notch pathway has been linked to the ephrin reverse signaling through the increase in EphrinB1, which altogether promoted stem-like features and pulmonary metastasis formation [120]. Additionally, increased expression of the cell migration-inducing protein was correlated with poor prognosis in osteosarcoma patient tissues and mechanistically related to Notch, as its specific genetic silencing suppressed expression and activation of Notch/Jagged1/HES1 signaling pathway in vitro and in vivo [121].

### 3.2. Hedgehog Signaling

The Hedgehog pathway is pointed to as a key regulator of embryonic development, exerting the control of cellular differentiation, proliferation, and self-renewal. Hedgehog is involved as well in the homeostatic regulation and stem cell renewal in adult somatic cells and tissues [122]. Hedgehog is a signaling cascade dependent on ligand–receptor interactions, which culminate in the activation of GLI transcription factors. In fact, GLI1 expression is the bona-fide readout of Hedgehog-activated status (Figure 4). Hedgehog target genes are molecules involved in the autocrine regulation of Hedgehog itself (e.g., *GLI1*, *PTCH1)* and other cell-specific genes regulating cell apoptosis, proliferation, and vascularization (e.g., *CYCLIND*, *MYC*, *BMI1*, *BCL2*, *VEGF*, and *SNAIL*) depending on the cell type and microenvironmental status [123].

The Hedgehog pathway is implicated in drug resistance and is constitutively active in a number of cancer types. This involvement appears to be mediated through paracrine signaling, interactions between the stroma and the tumor cells, and pathway activation in CSCs. In gastric cancer, doxorubicin-resistant cells that overexpress SHH and GLI1, the well-known signaling molecules involved in drug resistance, are susceptible to GLI inhibition with GANT61 and vismodegib via downregulation of its downstream effector *ABCG2* [124]. SMO antagonists (cyclopamine and sonidegib) decreased tumor burden in vivo and sensitized ALDH-positive cells and resistant ovarian cancer cells to paclitaxel [125]. GLI2 expression in patients with advanced non-small-cell lung cancer was shown to be more frequently positive in these patients than in those with controlled disease. This finding strengthens the association between the Hedgehog pathway activation and chemotherapy resistance in the clinical setting [126]. The Hedgehog pathway has been effectively linked to the self-renewal of CSCs in diverse tumors, such as glioblastoma [127], lung [128], and liver cancer [129].

#### Hedgehog Signaling in Osteosarcoma

Hedgehog pathway activation has been observed in osteosarcoma [130], and worse clinical outcomes are associated with overexpression of pathway components such as GLI2 [131]. Hirotsu and colleagues demonstrated that SMO genetic depletion or pharmacological inhibition with cyclopamine, a specific inhibitor of SMO, prevented cell proliferation in vitro and in vivo. This study discovered that a number of cell lines and biopsy samples overexpressed signaling molecules such as SMO, PTCH1, and GLI, suggesting that the pathway was activated in osteosarcoma [132]. The significance of GLI2 in human osteosarcoma has been extensively investigated in studies from the Setoguchi lab. This team showed in 2011 that GLI2 was overexpressed in biopsy tissues and that *GLI2* knockdown hindered in vitro cell proliferation by causing cell cycle arrest and down-regulating cell cycle promoters, namely cyclin D1 and phosphorylated Rb proteins [133]. Additionally, GLI2 appears to play a role in the spread of metastatic disease because its pharmacological inhibition with arsenic trioxide, vismodegib, or GANT61 inhibited cell invasion and migration while reducing lung metastasis [134]. Studies conducted later revealed that ribosomal protein S3 was responsible for regulating osteosarcoma cell invasion caused by GLI2. The forced expression of ribosomal protein S3 reversed the effects of GLI2 knockdown on this protein’s expression and cell migration. Since the expression of ribosomal protein S3 was higher in osteosarcomas with lung metastases compared to specimens that were not disseminated, this study suggested this signaling axis as a new marker of invasive osteosarcoma [135]. More recently, this group also reported that when combined with conventional chemotherapeutics, arsenic trioxide and vismodegib synergistically reduced cell proliferation. This treatment combination was also effective in a mouse xenograft model, establishing Hedgehog pathway inhibitors as an appealing therapeutic option for osteosarcoma [136].

More recent studies have shown that Hedgehog signaling can cooperate with other pathways to promote osteosarcoma aggressiveness [137] and is responsive to inhibition by natural compounds such as degalactotigonin through GSK3-β inactivation, leading to the diminished migration, invasion, and metastatic ability of tumor cells [138]. Interestingly, activation of Hedgehog in MSCs cooperates with Wnt/β-catenin to induce a pro-tumorigenic phenotype, including cartilage and bone tumor formation [139]. Moreover, Hedgehog is involved in CSC properties and chemoresistance [140]. We have previously demonstrated an inhibitory effect of the tankyrase inhibitor IWR-1 on Hedgehog transcriptional players such as *GLI2*, *PTCH1,* and *SMO* [141]. Unfortunately, few studies specifically explored this pathway in osteosarcoma CSCs as a mainstream therapeutic target.

### 3.3. Wnt/β-Catenin Signaling Pathway

Several signaling pathways control stem cell self-renewal, but Wnt/β-catenin is likely the most significant and researched, as reviewed by Steinhart and Angers [142]. The Wnt/β-catenin signaling is constitutively inactivated in normal bone cells and also in well-differentiated cancer cells (Figure 5). This pathway involves the interaction of a signal-receiving cell with secreted glycoproteins (ligands), which are lipid-modified by porcupine O-acyltransferases in a palmitoylation process. Canonical Wnt ligands such as Wnt2, Wnt3, Wnt3a, and Wnt8a may be sequestered by antagonists such as Wnt inhibitory factor 1 (WIF1) or Secreted frizzled-related protein, which silence the signaling pathway. Additionally, the interaction of LRP5/6 co-receptors with extracellular antagonists such as DKK-1 and Sclerostin directly inhibits Wnt activation. Additionally, the main signaling protein β-catenin is normally phosphorylated, ubiquitinated, and transported to proteasomal degradation at the cytoplasmic level (off state). This process is regulated by the β-catenin destruction complex, a multi-protein molecular structure made up of APC, casein kinase 1α, Axin 1/2, GSK-3β, and tankyrase proteins [142]. However, Wnt/β-catenin can be abnormally activated in cancer, in part due to the complexity of the pathway involving a large number of signaling molecules, but also due to mutations on key components such as APC. Wnt/β-catenin is implicated in epithelial and mesenchymal tumors, with special emphasis on leukemia, colon, breast, and prostate cancers, but also in bone tumors, as reviewed previously [143]. There is, therefore, a great interest and potential for the development of drugs targeting the Wnt pathway, which are currently under development or clinically approved and being repurposed for use in a cancer setting (Table 2). Drugs targeting Wnt are categorized, among others, in antibody-based treatments, derivatives of vitamin D, small molecule inhibitors, and non-steroidal anti-inflammatory drugs (NSAIDs) [144,145].

#### Wnt/β-Catenin Signaling in Osteosarcoma

Abnormal activation of Wnt/β-catenin has been related to the development of numerous carcinomas and linked to CSC self-renewal in diverse solid tumors, including osteosarcoma [143]. However, reports on the level of Wnt/β-catenin activation in osteosarcoma are conflicting, and no clear causal link has yet been found.

Based on the discovery of Wnt ligands, LRP5/6 co-receptors, or cytoplasmic β-catenin staining, some investigators claimed that osteosarcoma samples had aberrant Wnt/β-catenin activation mediated by an autocrine mechanism. A metastatic phenotype in osteosarcoma has been linked to elevated β-catenin levels [152]. For example, osteosarcoma tumors overexpressing LRP5, a Wnt co-receptor, and nuclear β-catenin are associated with a poorer prognosis and decreased patient survival [153]. Another study made the same claim, arguing that lung metastasis formation is caused by aberrant Wnt/β-catenin signaling activity [154]. Furthermore, high levels of β-Catenin were linked to the development of osteosarcoma lung metastasis and osteoprogenitor proliferation [155,156]. Additionally, Vijayakumar et al. found that Wnt signaling was active in half of the human sarcomas and cell lines analyzed [157]. Rubin et al. showed that the re-expression of the WIF-1 antagonist decreased tumor development and metastasis in osteosarcoma animal models by inhibiting Wnt signaling [158]. Therefore, by impeding terminal osteogenic differentiation and encouraging cell proliferation, dysregulation of the Wnt signaling pathway may contribute to osteosarcoma cell aggressiveness. These results suggest that disrupting the Wnt-pathway could be a promising therapeutic approach for treating metastatic OS.

Other studies that analyzed nuclear β-catenin accumulation rather than cytoplasmic β-catenin accumulation demonstrated a down-regulation of Wnt/β-catenin in osteosarcoma biopsy samples and osteoblastoma in comparison to normal osteoblasts. In 90% of the analyzed biopsies and cell lines, β-catenin was not present in the nucleus, and in the remaining cases, there was only faint nuclear staining. This study hypothesized that the loss of Wnt/β-catenin pathway activity leads to the development of osteosarcoma and underlined the significance of nuclear staining in determining the level of Wnt activity since transcription for target gene expression takes place in the nucleus [159]. Inactivation of Wnt was also previously observed, implying a tumor suppressive role for Wnt/β-catenin signaling [160]. Additionally, inactivation of the Wnt antagonist, *WIF1*, has been related to radiation-induced osteosarcoma in mice, further suggesting the deregulation of Wnt signaling in osteosarcomas [152].

Targeting the Wnt-pathway could be a potential strategy for therapy, as it seems that its abnormal activation causes the transcription of oncogenes and cell cycle promoters, resulting in increased cell proliferation and survival (Figure 5) [145]. Moreover, some recent studies demonstrate that activation of Wnt is related to both the growth and invasion of osteosarcoma cells by long non-coding RNAs such as *HNF1A-AS1* [161], *AWPPH* [162], *MRPL23-AS1* [163], and *SNHG10* [164]. The role of these non-coding RNAs [165] and also of microRNAs [166] and their relation with CSCs in osteosarcoma were recently analyzed.

Several studies already reported results targeting the diverse Wnt pathway players in osteosarcoma [167]. Goldstein et al. used the monoclonal antibody BHQ880 to inhibit Dkk-1 in tumor-bearing mice and showed that the serum levels of this Wnt antagonist are higher in rapidly growing tumors than in large tumors displaying reduced growth rates. BHQ880 slowed the growth of tumors and also of metastasis formation, which was paralleled by overall activation of canonical Wnt components and increased cell differentiation as assessed by expression of the bone marker osteopontin [168]. This finding is in line with other studies that reported abnormally high levels of Dkk-1 at the frontline periphery of excised tumor biopsies displaying high mitotic rates and rapid bone remodeling [169]. BHQ880 has, in fact, a promising therapeutic role also in multiple myeloma as it reduces the formation of osteolytic lesions [170]. Nevertheless, the modulation of Dkk-1 should consider its basal serum levels, but also transcriptional regulation [171] and the crosstalk with non-canonical Wnt pathways [172]. When targeting the Wnt-pathway, activating mutations in downstream molecules, for example, *CTNB1*, can be of a negative influence as it may bypass Wnt inhibition and preserve the invasive phenotype of the cells [173]. A pre-clinical investigation by Leow et al. demonstrated that inhibiting the Wnt/β-catenin pathway with curcumin and PKF118-310 reduced nuclear β-catenin levels, which in turn reduced intrinsic and activated β-catenin/TCF transcriptional activity and, consequently, the expression of β-catenin target genes. This resulted in the down-regulation of MMP-9, a reduction in the expression of cyclin-D, c-MYC, and survivin, and inhibition of the potential for migration. This had a suppressive effect on cell proliferation and increased cell mortality [148]. It also seems that blocking the Wnt/Snail axis can reduce tumor invasiveness by EMT reversal [150], while other authors report that the Wnt/β-catenin pathway was activated during the EMT of osteosarcoma driven by BMP-2 [174].

From the abovementioned studies, we infer that the activation status and strategies proposed for osteosarcoma treatments aiming to target Wnt/β-catenin are conflicting to some extent, which may be explained by the few studies before 2015–2016 that specifically addressed the role of this pathway in isolated CSCs in osteosarcoma. The majority of them focused solely on cell lines and human tumor samples, taking into account bulk tumor cells, which prevents conclusions about the regulatory role of Wnt/β-catenin signaling in CSCs because they are thought to make up a tiny portion of the total cell population. However, given the critical role of Wnt signaling in modulating the delicate balance between self-renewal and differentiation also in MSCs [175], as well as in controlling the stemness networks in other adult stem cells, points to dysregulation of this developmental route in osteosarcoma CSCs. In fact, our previous studies revealed a specific activation of Wnt/β-catenin signaling in osteosarcoma stem-like spheres, based on nuclear β-catenin accumulation, expression of target genes such as *AXIN2,* and also enhanced TCF/LEF transcriptional activation [176]. The contribution of this pathway to osteosarcoma stemness was further substantiated in our subsequent study, in which we demonstrated that Wnt/β-catenin inhibition with the tankyrase inhibitor IWR-1 reversed the doxorubicin-induced Wnt-activation and acquisition of stemness features, such as expression of pluripotency-related and drug efflux-related markers, in differentiated osteosarcoma cells [177]. Moreover, we further demonstrated that IWR-1 attenuated Wnt/β-catenin signaling specifically in osteosarcoma CSCs and inhibited in vivo the growth of a subcutaneous xenograft [141]. Liu et al. proposed that the natural compound dioscin inhibits CSCs and tumor growth of osteosarcoma through the repression of Akt/GSK3/β-catenin [178].

In very recent years, an increasing number of research groups have been interested in the detrimental role of canonical Wnt, specifically in the osteosarcoma CSC subsets. Zhao et al. investigated how the stress response protein NDRG1, which inhibits multiple oncogenic pathways, affects mitochondria and CSCs. They discovered that NDRG primarily functions via the Wnt pathway, as demonstrated by its ability to modulate *WNT3A* expression and the differentiation of osteosarcoma CD133^+^ CSCs by downregulation of pluripotency factors [179]. Another signaling pathway also involved with canonical Wnt in osteosarcoma CSCs is histone methyltransferase SETD2, whose overexpression enhances sensitivity to cisplatin by suppressing Wnt/β-catenin signaling and its downstream target genes *CMYC*, *CD133*, and *CCND1,* due to β-catenin degradation mediated by H3K36me3 modification in *GSK3B* loci [180]. As for other self-renewal pathways, Wnt/β-catenin seems to be modulated by pluripotency factors, such as SOX-2, which, when genetically depleted, leads to downregulation of Wnt-related proteins [181] and also by long non-coding RNAs such as *DLX6-AS1* [182]. Further studies focused on osteosarcoma CSC sub-populations are required to fully understand the promising therapeutic potential of Wnt/β-catenin signaling in this aggressive bone tumor.

## 4. Pluripotency Transcription Factors in Cancer Cells and Interconnections with Drug Resistance

Virchow proposed that cancers have characteristics similar to those of embryonic cells and speculated that tumors arise from embryo-like cells after a pathological examination that revealed a high degree of cellular heterogeneity [183]. Others then postulated that tumors could develop from dormant embryonic remnants found in adult tissues that, when awakened under specific circumstances, could give rise to malignancies [184,185].

In the last ten years, there has been a significant increase in research on the fundamental significance of embryonic factors in driving tumorigenesis and reprogramming cancer cells toward a stem-like phenotype. Embryonic stem cell (ESC) transcription factors such as NANOG, Oct-4, KLF4, and SOX-2 are reported to be involved in the stemness features of cancer cells, including in the process of dormancy/reactivation induced by chemotherapy [186]. Expression of, e.g., Oct-4 inversely correlates with clinical prognosis and chemoresistance in some tumors, but not all studies report a prognostic role of this marker [187]. These markers cooperate to confer pluripotency characteristics and can bind to each other to stimulate nuclear translocation, although they have also demonstrated cytoplasmic localization in a variety of cancers [188]. In general, the expression of ESC markers in tumor cells contributes to tumor aggressiveness, including the induction of drug resistance-related mechanisms [189].

Yamanaka’s group performed groundbreaking experiments in 2006 that showed how to force the expression of a core of embryonic transcription reprogramming factors (Oct-3, SOX-2, KLF4, and c-MYC), also known as OSKM or Yamanaka factors, to transform fibroblasts, differentiated mature cells, into induced pluripotent stem cells [190]. This hypothesis was later applied to a cancer cell setting as demonstrated by several studies, e.g., in epithelial CSCs and neuroblastoma [191,192], although others have shown that, for instance, SOX-2 is not crucial for the reprogramming of primary mouse melanocytes and melanoma cells into induced pluripotent stem cells [193], neither for primary tumor formation nor metastatic spreading [194].

Mechanistically, NANOG interacts with and up-regulates HDAC1 to promote multidrug resistance and stemness features in immune-edited cervical cancer cells [195]. In melanoma cells, *POU5F1* expression or transmembrane delivery of Oct-4 protein induced the dedifferentiation and acquisition of typical CSC-like features, including increased expression of KLF4 and NANOG [196]. In head and neck squamous carcinoma cells, *POU5F1* expression promoted tumor growth, resistance to cisplatin through *ABCC6* expression, and in vivo tumorigenicity, and was correlated with poor histologic grade [197]. The transcription of *POU5F1* can be suppressed by the tryptophan metabolite, ITE, via a mechanism involving the aryl hydrocarbon receptor [198], and the combination with a specific AKT inhibitor decreased the proliferation of embryonal carcinoma cells, adherent cancer cells, and CSC-enriched spheres [199].

Stolzenburg et al.’s elegant study showed that selective SOX-2 targeting using a zinc finger-based synthetic transcription factor lowered mRNA expression in breast cancer cells and led to decreased cell proliferation in vitro and in vivo. This research revealed that SOX-2 is possibly a “druggable” molecule and, therefore, a therapeutic target [200]. *SOX2* knockdown in adenocarcinoma cells enhanced sensitivity to cisplatin via Wnt-β-catenin inhibition and increased apoptotic cell death [201], with its expression and contribution to self-renewal being also decreased via HDAC11 inhibition mediated by Gli1 [202]. In melanoma, modulation of *SOX2* expression was demonstrated to be crucial for the existence of side-population cells and to induce *ABCC1* expression that conferred resistance to paclitaxel [203]. A new model system to isolate stem-like cancer cells was recently developed based on the functional transcriptional activity of SOX-2. SOX-2-high cells were more metabolically active, proliferative, migratory, invasive, and drug-resistant [204].

KLF4 expression varies by cancer type, and during carcinogenesis, it may be tumor-suppressive or oncogenic, depending on the tumor stage. For instance, in vivo models of colonic carcinogenesis revealed the tumor suppressive function of KLF4, which is typically down-regulated in colorectal cancer. This research demonstrated that KLF4 is crucial for limiting the growth of colonic neoplasia and is involved in acinar-to-ductal cell reprogramming [205]. Additionally, KLF4 is up-regulated in early pancreatic carcinogenesis and is necessary for acinar-to-ductal metaplasia; its ablation lessens the development of pancreatic intraepithelial neoplasia caused by mutant *KRAS* [206]. However, for instance, in esophageal squamous cell carcinoma, *KLF4* expression is decreased, and its deletion produces squamous cell dysplasia in mouse models. *KLF4* expression is downregulated in high-grade dysplasia and early esophageal squamous cell cancers, but it increases with advanced cancer stage, and it is negatively connected with survival in these malignancies [207]. Moreover, subcellular localization of KLF4 influences non-small cell lung cancer cells’ resistance to cisplatin, and its nuclear expression was related to the histological grade and clinical stage of patients [208].

The expression of embryonic transcription factors as a tool for diagnostic or therapeutic strategies should be used carefully considering the cellular context and tumor type. In fact, manipulating embryonic and pluripotency-related transcription factors may demonstrate therapeutic efficacy for tumor cells but may, unfortunately, hamper key signaling cascades in normal somatic stem cells. This is of special concern considering the progress of the experimental technologies (e.g., CRISPR) to edit the genomic and epigenomic cellular landscapes [209] and the knowledge of the mechanisms involved in the cellular response to pharmacologic interventions due to the manipulation of complex gene expression patterns [210].

### Molecular Hallmarks of Pluripotency Modulate Osteosarcoma Stemness

During the last five years, several reports explored the prominent role of embryonic transcription factors in osteosarcoma CSCs. These are involved in the proliferation and chemoresistance of CSCs to the common therapeutics used in osteosarcoma, such as doxorubicin, and are also induced by drugs in bulk non-stem osteosarcoma cells.

SOX-2 has been the most well-studied embryonic transcription factor in osteosarcoma [176], with its role in tumor initiation and progression being well characterized. Some studies specifically investigated SOX-2 in osteosarcoma CSCs and its crosstalk with other signaling pathways. Verma et al. uncovered a feedback mechanism by which SOX-2 and YAP1 regulate each other’s expression, and both are crucial to osteosarcoma CSCs maintenance [211]. In fact, conditional knockout of *SOX2* expression was demonstrated to drastically reduce the onset of tumor formation in a mouse model of osteosarcoma [212]. Some microRNAs have also been shown to modulate SOX-2 expression. *MIR-34a* seems to be involved in the dedifferentiation of osteosarcoma via plasminogen activator inhibitor-1, whose inhibition suppressed the upregulation of SOX-2 [213]. miR-21-5p inhibition downregulated SOX-2, Oct-4, and NANOG proteins mediated by inactivation of Wnt/β-catenin [214]. Additionally, *MIR-429* directly targets *SOX2* and transcriptionally regulates its expression through *KLF8* [215]. Other authors have shown that SOX-2 expression might be molecularly targeted by drugging proteins, such as the deubiquitinase USP9x with neogambogic acid [216] and STAT3 signaling with apatinib [217], which then leads to inhibition of stemness features, including expression of Oct-4 and NANOG, and increase chemosensitivity in osteosarcoma cells.

It is known that KLF4 may act as an oncogene in osteosarcoma [218] by mechanisms involving Wnt/β-catenin, Notch, and EMT, despite that it seems to not have a relevant prognostic role in some tumors [219]. Nevertheless, several authors explored the contribution of KLF4 to osteosarcoma stemness. The expression of this ESC transcription factor may be downregulated by miR-10b [220], and also miR-135a reduces pulmonary metastasis by targeting BMI1 and KLF4 [221]. MG-63 cells retrovirally transduced with *OCT3/4*, *KLF4*, and *SOX2* genes demonstrated enhanced CSC properties, namely expression of CD24, CD26, and CD133, chemoresistance, cell migration, and tumorigenesis, and capacity to differentiate into osteogenic cells [222]. Moreover, Zhang et al. demonstrated an elevated expression of KLF4 in patient-derived tissues compared to normal tissues, which correlated with increased cell proliferation and metastasis via the upregulation of the molecular chaperone CRYAB [223]. However, inhibition of KLF4 with simvastatin reversed the stemness features and tumorigenesis induced by doxorubicin in vivo [224].

More recently, several studies demonstrated that expression of ESC transcription factors in osteosarcoma cells, specially NANOG, is decreased by compounds such as dimethylaminomicheliolide [225], ethanoic C. cassia extracts [226], and also the anesthetic levobupivacaine [227]. Others demonstrated that Oct-4 promotes osteosarcoma by modulating the expression of the long non-coding RNA *AK055347* [228].

The diverse studies exploring the dependency of stemness maintenance in osteosarcoma cells on the expression of embryonic factors clearly demonstrate the benefits of a potential inhibitory therapeutic strategy. However, more research is needed, especially using in vivo studies and human-derived tissues to a better comprehension of the leading oncogenic role of ESC factors in both osteosarcomagenesis and chemotherapeutic resistance.

## 5. Conclusions

The curing of high-grade osteosarcoma patients has been delayed by the worrying rates of tumor local recurrence and metastatic disease, likely due to poor response to standard therapy. Resistant CSCs play a leading role in the unsatisfactory therapeutic outcomes, besides the clear improvement in the research of new therapeutic targets and compounds. Based on extensive biological research during recent decades, the existence of CSCs with distinctive properties of clonogenicity, chemoresistance, self-renewal, and tumorigenicity, within an individual tumor, is no longer questionable. However, therapies directed to CSC have been difficult to establish for osteosarcoma treatment, as well as for other malignancies. Several reasons may account for this delayed translational application of preclinically developed compounds to a robust clinical testing, namely, technical issues concerning the isolation of the rare CSCs, difficulties in identifying specific surface markers that can substantiate the stem cell-like phenotype in osteosarcoma (for instance, CD133, CD29, CD90, CD105, CD44, ICAM-1, CD56, CD117, CBX3/ABCA5, CD248, CD271, CD49b, and CD24 are also expressed by several other tumors) and the histological validation in patient clinical samples, as we previously extensively discussed [8]. In addition, the identification of individual markers that may be used as therapeutic targets is hampered by the complex genetic and microenvironmental landscape underlying osteosarcomagenesis, as we discussed above in this review.

Distinguishing univocally the pathways and molecular markers that CSCs use to bypass conventional therapies of osteosarcoma is detrimental to creating CSC-targeted therapies. However, additional studies, especially using patient-derived tumor specimens, are necessary for the field so that we can better understand the differences and similarities between osteosarcoma CSCs and normal stem cells, namely MSCs. Of special note, it is also of interest to discover molecular targets dissimilar from circulating stem/progenitor cells of the hematopoietic system, which may also reside in the bone marrow niche. To achieve this aim, the field of molecular targeting of CSCs may benefit from the innovative approaches for cancer treatment developed in recent years, such as nanotheranostics [229,230], lipid raft modulation [231], thermal ablation, magnetic hyperthermia, radiomics, and pathomics, as recently summarized [232,233]. These strategies may help bypass the side effects caused by conventional therapies and will certainly help to refine the postulated CSC-targeted therapies. Nevertheless, evaluating the long-term pharmacodynamic effects of those recent strategies and also of emerging repurposed drugs in normal stem cell compartments is fundamental to assure their clinical efficacy and safety. Considering the small numbers of these cells in both tumor and normal tissues, it is imperative to trace their course after these therapies are applied. The keen technological developments in the field of single-cell analysis [234] will definitely contribute to elucidating the molecular mechanisms underlying tumor cell response to therapy.

## Figures and Tables

**Figure 1 ijms-24-08401-f001:**
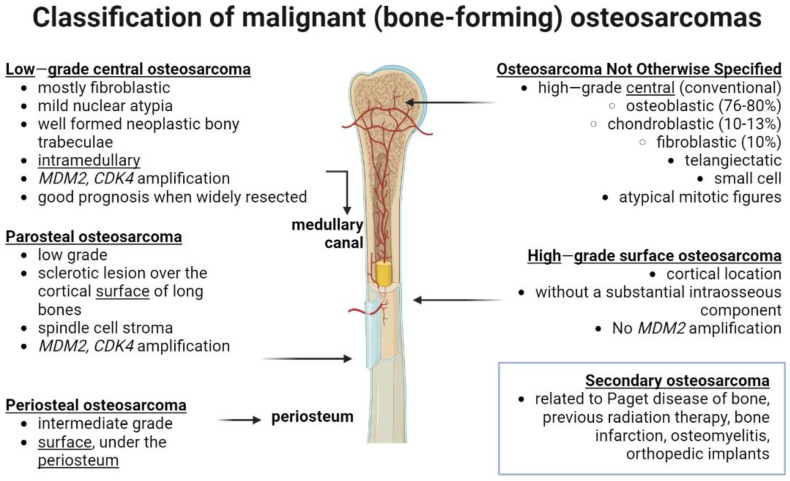
Updated classification of human malignant, bone mass-depositing osteosarcoma considering the latest WHO 2020 guidelines for bone tumors. The picture outlines the location of tumors (note that it can be central or in the surface of the bone, or intramedullary in the bone marrow medulla) and associated molecular and clinical features. Compiled from revision of literature from [34,37,38]. *MDM2*— human homolog of mouse double minute 2; *CDK4*—cyclin-dependent kinase 4. Created with BioRender.com (figure created on 13 March 2023).

**Figure 2 ijms-24-08401-f002:**
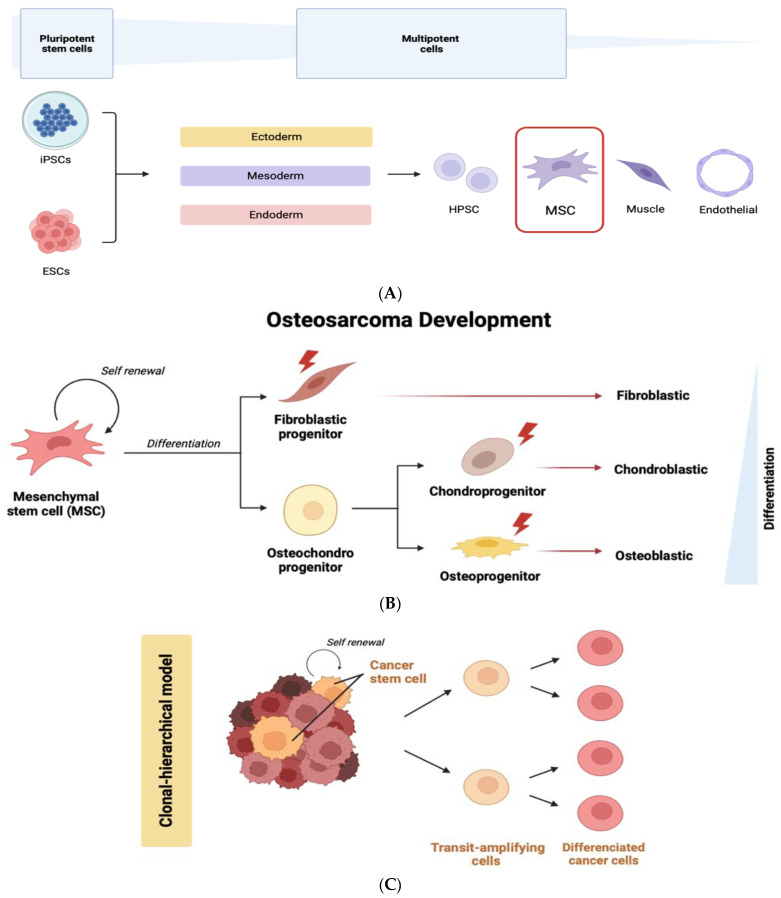
(**A**) Development potential of iPSCs and ESCs into ectoderm, mesoderm, and endoderm multipotent cells, highlighting the formation of MSCs. (**B**) Osteosarcoma development may result from an abnormal differentiation of MSCs. Mutations occurring in osteoblastic, chondrogenic, or fibroblastic ancestor cells may originate genetically altered cells that contribute to bone sarcoma formation. (**C**) In a clonal-hierarchical model, new mutations occurring in CSCs may lead to clonal evolution, and multiple CSC clones may co-exist within the tumor, which altogether constitutes a complex tumor heterogeneity. iPSCs—induced pluripotent stem cells, ESCs—embryonic stem cells, HPSCs—hematopoietic progenitor and stem cells, and MSCs—mesenchymal stem cells. Created with BioRender.com (figure created on 28 March 2023).

**Figure 3 ijms-24-08401-f003:**
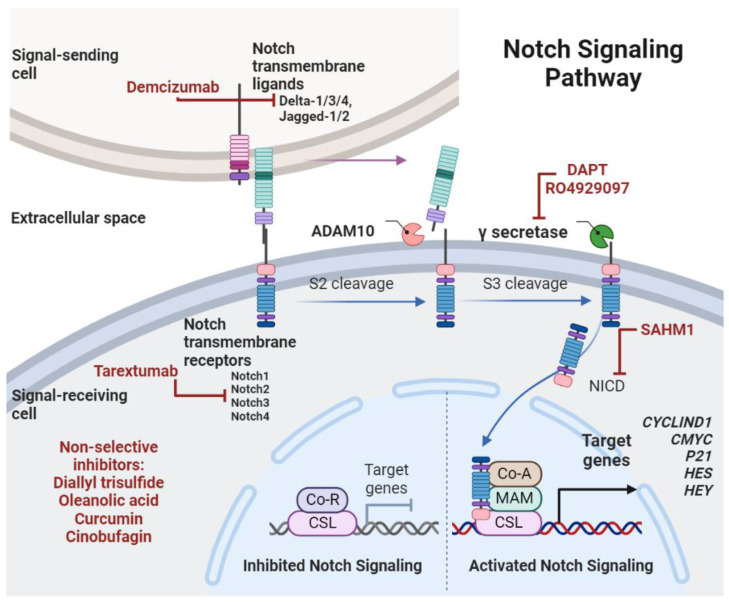
Notch ligands interact with transmembrane receptors and activate a signaling cascade that leads to the transcriptional activation of Notch-associated target genes (e.g., *CMYC, P21*). Drugs and natural compounds that inhibit molecular players of Notch signaling cascade are shown in red, and most of them were already tested in pre-clinical studies in osteosarcoma. Abbreviations: ADAM10—a disintegrin and metalloproteinase domain-containing protein 10, NICD—Notch intracellular domain, Co-R—co-repressor protein, CSL—“CBF1/Su(H)/LAG-1” DNA-binding protein complex, Co-A—co-activator protein, MAM—Mastermind protein, *HES*—Hairy and enhancer of split, *HEY*—Hairy/enhancer-of-split related with YRPW motif protein, DAPT—N-[N-(3, 5-difluorophenacetyl)-l-alanyl]-s-phenylglycinet-butyl ester, SAHM1—stapled α-helical peptide derived from mastermind-like 1. Created with BioRender.com (figure created on 31 March 2023).

**Figure 4 ijms-24-08401-f004:**
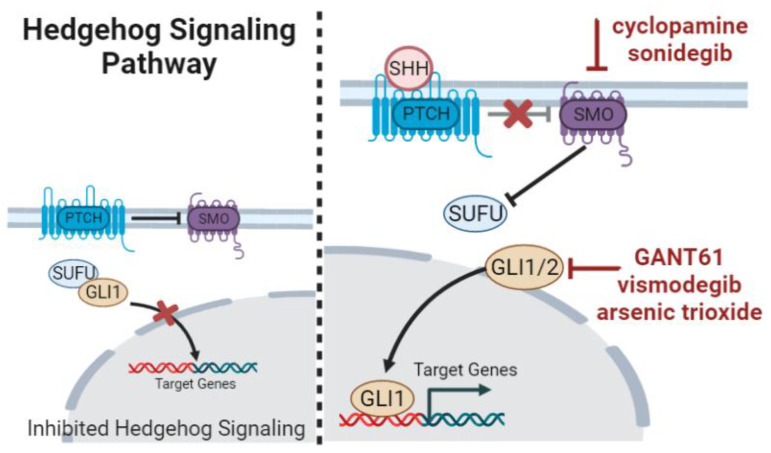
Overview of the ligands, receptors, and transcriptional regulators of Hedgehog signaling pathway. Some of the drugs and natural compounds discussed in this text, which inhibit molecular players of this signaling cascade, are shown in red. PTCH—patched, SMO—smoothened, SUFU—suppressor of fused, GLI—Glioma-associated oncogene homolog, and SHH—Sonic hedgehog ligand. Created with BioRender.com (figure created on 14 March 2023).

**Figure 5 ijms-24-08401-f005:**
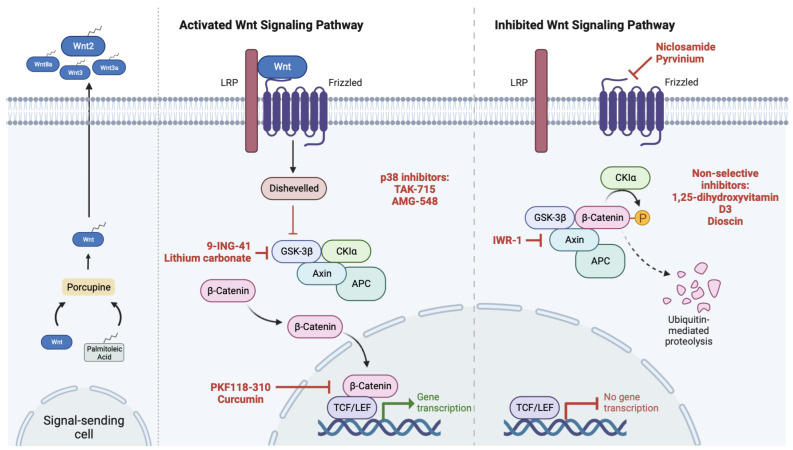
Overview of the key components involved in the regulation of activation/inactivation of the canonical or Wnt/β-catenin pathway. Some of the drugs and natural compounds that inhibit molecular players of the Wnt/β-catenin signaling cascade are shown in red. These drugs have been tested in preclinical studies, as outlined in the main text, are in clinical trials, or are repurposed approved drugs tested in osteosarcoma with the aim of targeting Wnt pathway (see also Table 2). LRP—Low-density lipoprotein receptor-related protein, GSK-3β—Glycogen synthase kinase 3β, CKIα—Casein kinase 1α, APC—Adenomatous polyposis coli, and TCF/LEF—T-cell-specific transcription factor/Lymphoid enhancer-binding factor. Created with BioRender.com (figure created on 27 April 2023).

**Table 1 ijms-24-08401-t001:** Summary of some of the genetic alterations observed in osteosarcoma tumor tissues and related effects on osteosarcoma development.

Genetic Alteration	Effects on Osteosarcoma Development	References
*TP53* and *RB1*	Tumor suppressor proteins and regulators of cell cycle, whose deregulation leads to elevated genomic instability; mutations predict poor survival; inability of defective RB protein to block the G1 to S transition	[46,47,48]
*MDM2* and *CDK4*	MDM2 inhibits p53 and targets p53 for proteasomal degradation; their co-expression in high-grade tumors suggests their progression from low-grade tumors; they promote osteosarcoma cell proliferation	[49,50]
*CDKN2A*	Promotes the stability of the cellular genome; tumor suppressor inactivated in RB wild-type osteosarcoma; loss of expression is predictive of poor prognosis	[51,52]
*PTEN*	Chromosomal losses facilitate osteosarcoma cell proliferation	[53,54]
*TWIST* and *MET*	Genomic deletions related to alterations also in CKIT and APC; correlation with poor outcomes	[55,56]
*CMYC*	Driver gene for osteosarcomagenesis; promotes cell invasion via MEK-ERK activation	[45,57]
*FGFR1*, *FGFR2*, *FGFR3*	Amplification, correlation with poor response to therapy	[58,59]
Chromothripsis	Fragmentation and rearrangement of chromosomal regions leading to amplification, gain, or disruption of oncogenic driver-genes; correlation with clinical outcome	[45,60]
Kataegis	Hypermutation pattern, including mutations in TP53, RB1, ATRX, and DLG2	[61]
Alternative lengthening of telomeres	Most high-grade tumors maintain their telomeres during cell division using this mechanism	[62]

**Table 2 ijms-24-08401-t002:** List of new drugs in clinical trials enrolling osteosarcoma patients and repurposed approved drugs in preclinical tests in osteosarcoma, whose mechanisms of action target Wnt pathway components.

Drugs Targeting Wnt Players in Clinical Trials, Enrolling Osteosarcoma Patients	Study Phase/Mechanism of Action on Wnt Pathway	ClinicalTrials.Gov Identifier NCT Number
9-ING-41 with doxorubicin	Phase I/II/Targeting GSK-3β	NCT03678883, [146]
Lithium carbonate combined with neo-adjuvant chemotherapy	Phase IV/Targeting GSK-3β	NCT01669369, [146]
**Drugs in Clinical Use Repurposed in Preclinical Tests in Osteosarcoma**	**Original Application/Potential Mechanism on Wnt Pathway**	**References**
TAK-715 and AMG-548	p38 inhibitor/inhibits Casein Kinase 1δ, ε in U2OS cells	[147]
Curcumin and PKF118-310	Natural products/reduces nuclear β-catenin in U2OS, SaOS-2, and HOS cells	[148]
Niclosamide and Pyrvinium	Anthelmintic drug/promotes Fzd1 endocytosis; inhibits the Wnt/Axin-2/Snail axis	[149,150]
1,25-dihydroxyvitamin D_3_	Acts at different levels/antagonistic action on activated Wnt or β-catenin	[151]

## Data Availability

No new data were created or analyzed in this study. Data sharing is not applicable to this article.

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
