# Peer review of "Self-Renewal and Pluripotency in Osteosarcoma Stem Cells’ Chemoresistance: Notch, Hedgehog, and Wnt/β-Catenin Interplay with Embryonic Markers"

_ijms, 2023, doi:10.3390/ijms24098401_

Round 1

Reviewer 1 Report

This is a thorough review of osteosarcoma stem cells, an important area to review.

I don’t think the classification of osteosarcomas fits within this review. Same with section 2.3. The review is sufficiently long and would benefit from focusing on the stem cells.

Discuss the studies of DKK1 antibodies in osteosarcoma such as PMID: 27049730.

How were the drugs in Figure 5 chosen? The legend says “most of them were already tested in pre-clinical studies in osteosarcoma”. You should choose those that have been tested in osteosarcoma or those that are in clinical trials. Be complete and consistent. For example PRI-724 has been tested in pre-clinical studies and is in clinical trials for solid tumors.

A section on osteosarcoma stem cell markers would be helpful. A few are mentioned throughout without much context.

Minor comments:

“Revised” is used wrong on line 70 and 518. I think you mean “reviewed”.

Define abbreviations in legends such as in Fig. 3: co-R and MAM

Line 412 uses “LDE225” but the figure has sonidegib. Be consistent.

Figure 5 has PORCN and LGK974 without mention in the text. The figure should show secretion of WNT ligands.

Paragraph starting line 372: The first sentence says in vivo mouse models. But then the second sentence says in vitro. The flow of topic/ideas in this paragraph needs improvement.

Line 373: metastasization is not a word.

Line 470 you mention “drugs that are on the market”. Could you please list them with references?

Line 526: drug is spelled wrong.

The abbreviation “et al.” has a period.

Title wording is a bit off. Stem cells should be possessive.

Title needs fixing. Few minor errors

Author Response

Author's Reply to the Review Report (Reviewer 1)

This is a thorough review of osteosarcoma stem cells, an important area to review.

We are very thankful to the compliment and insightful comments from the reviewer.

  1. I don’t think the classification of osteosarcomas fits within this review. Same with section 2.3. The review is sufficiently long and would benefit from focusing on the stem cells.

We included a section regarding the classification of osteosarcoma human tumors since there was a recent update from the World Health Organization and only few studies reported a concise summary of the new classification. We also reformulated some sentences, also in line with the suggestion of reviewer 2.

Regarding section 2.3 we shortened the paragraphs, and in line with the request from reviewer 2, we added a new table 1, summarizing the effects of genetic alterations in osteosarcoma development.

  1. Discuss the studies of DKK1 antibodies in osteosarcoma such as PMID: 27049730.

As suggested by the reviewer, we added a paragraph discussing some studies regarding DKK-1 modulation in osteosarcoma, in section 3.3.1.

  1. How were the drugs in Figure 5 chosen? The legend says “most of them were already tested in pre-clinical studies in osteosarcoma”. You should choose those that have been tested in osteosarcoma or those that are in clinical trials. Be complete and consistent. For example PRI-724 has been tested in pre-clinical studies and is in clinical trials for solid tumors.

As suggested by the reviewer, we revised extensively the information contained in Figure 5. Since the reviewer also suggested that we listed the drugs targeting Wnt that are on the market, we then include a new Table 2 with the two clinical trials enrolling osteosarcoma patients which test Wnt modulators, and information about those clinically-approved and repurposed drugs that are being tested pre-clinically in osteosarcoma. Based on this, Figure 5 shows in red the drugs that are mentioned throughout the main text (e.g. IWRs) and those listed in Table 2.

  1. A section on osteosarcoma stem cell markers would be helpful. A few are mentioned throughout without much context.

We understand the concern of the reviewer regarding the osteosarcoma cancer stem cell markers. However, since we recently published another review article https://doi.org/10.3390/ijms231911416 and extensively discussed this subject on that article, we herein included a summarized mention to the surface markers in osteosarcoma in the conclusion section.

Minor comments:

  1. “Revised” is used wrong on line 70 and 518. I think you mean “reviewed”.

As recommended by the reviewer we checked the use of the word “revised” in the whole manuscript and substituted the word in several sentences, using instead for instance the words “reviewed”, “analyzed” and “summarized”.

  1. Define abbreviations in legends such as in Fig. 3: co-R and MAM.

As recommended by the reviewer we defined abbreviations in Fig. 3 and in the legends of all the figures of the manuscript.

  1. Line 412 uses “LDE225” but the figure has sonidegib. Be consistent.

As recommended by the reviewer we used sonidegib in the main text and in the corresponding figure 4.

  1. Figure 5 has PORCN and LGK974 without mention in the text. The figure should show secretion of WNT ligands.

We removed PORCN and LGK974 from the figure. As recommended by the reviewer we now added a signal-sending cell, showing the secretion of Wnt ligands, and updated the main text accordingly.

  1. Paragraph starting line 372: The first sentence says in vivo mouse models. But then the second sentence says in vitro. The flow of topic/ideas in this paragraph needs improvement.

As recommended by the reviewer we rephrased this paragraph.

  1. Line 373: metastasization is not a word.

As recommended by the reviewer we rephrased the sentence using “metastasis formation”.

  1. Line 470 you mention “drugs that are on the market”. Could you please list them with references?

As recommended by the reviewer, we included a table with the drugs that are approved in the market and in clinical use, which were repurposed and tested in preclinical studies in osteosarcoma to infer about their potential as Wnt target molecules – see Table 2.

  1. Line 526: drug is spelled wrong.

As recommended by the reviewer we corrected the word PKF118-310.

  1. The abbreviation “et al.” has a period.

As recommended by the reviewer we checked the abbreviation “et al.” in the whole text and added the missing periods.

  1. Title wording is a bit off. Stem cells should be possessive.

As suggested by the reviewer, we reformulated the title of this manuscript to “Self-Renewal and Pluripotency in Osteosarcoma Stem Cells’ Chemoresistance: Notch, Hedgehog and Wnt/β-Catenin interplay with Embryonic Markers”.

Reviewer 2 Report

The review discussed the major signalling pathways involved in the development of osteosarcoma, such as The Notch, Hedgehog, and Wnt/β-Catenin, and their potential use as therapeutic targets. It is a well-written and comprehensive review I only have a few comments:

1.      In multiple instances, the authors mentioned “in this issue of IJMs” (e.g. line 70, line 272).  These phrases should be removed to avoid potential biases in citing articles.

2.      Line 124: osteoid and bony matrix are also characteristics of normal bone. I suggest either removing the phrase “defective bone” or rephrasing it as osteosarcomas are… osteogenic tumours that produce defective bone with osteoid or bony matrix.

3.      For the other genetic alteration (CDK4, CDKN2A, PTEN, TWIST, CMYC, and FGFR2) as summarised in line 179, can the authors make a table to summarise their effects on osteosarcoma development?

4.      Line 574:  remove anciently. 

Good quality of english

Author Response

Author's Reply to the Review Report (Reviewer 2)

The review discussed the major signalling pathways involved in the development of osteosarcoma, such as The Notch, Hedgehog, and Wnt/β-Catenin, and their potential use as therapeutic targets. It is a well-written and comprehensive review I only have a few comments:

We are very thankful to the compliment and insightful comments from the reviewer.

  1. In multiple instances, the authors mentioned “in this issue of IJMS” (e.g. line 70, line 272). These phrases should be removed to avoid potential biases in citing articles.

As recommended by the reviewer we removed the mentions to IJMS in the main text.

  1. Line 124: osteoid and bony matrix are also characteristics of normal bone. I suggest either removing the phrase “defective bone” or rephrasing it as osteosarcomas are… osteogenic tumours that produce defective bone with osteoid or bony matrix.

As recommended by the reviewer we rephrased the sentence in this line.

  1. For the other genetic alterations (CDK4, CDKN2A, PTEN, TWIST, CMYC, and FGFR2) as summarised in line 179, can the authors make a table to summarise their effects on osteosarcoma development?

As suggested by the reviewer we included a table in the manuscript summarizing the effects of some genetic alterations in osteosarcoma development – “Table 1. Summary of some of the genetic alterations observed in osteosarcoma tumor tissues and related effects on osteosarcoma development.”

  1. Line 574: remove anciently.

As recommended by the reviewer we removed the word “anciently” in this line.